# Composite Material of PDMS with Interchangeable Transmittance: Study of Optical, Mechanical Properties and Wettability

Flaminio Sales [1,†], Andrews Souza [1,†], Ronaldo Ariati [1,†], Verônica Noronha [1,†], Elder Giovanetti [1], Rui Lima [2] and João Ribeiro [1,3,*]

1   ESTiG, Instituto Politécnico de Bragança, 5300-252 Bragança, Portugal; a42848@alunos.ipb.pt (F.S.); a38195@alunos.ipb.pt (A.S.); a46685@alunos.ipb.pt (R.A.); noronhaveronica@hotmail.com (V.N.); a42849@alunos.ipb.pt (E.G.)
2   MEtRICs, Mechanical Engineering Department, Campus de Azurém, University of Minho, 4800-058 Guimarães, Portugal; rl@dem.uminho.pt
3   CIMO, Instituto Politécnico de Bragança, 5300-252 Bragança, Portugal
*   Correspondence: jribeiro@ipb.pt
†   These authors contributed equally to this work.

**Abstract:** Polydimethylsiloxane (PDMS) is a polymer that has attracted the attention of researchers due to its unique properties such as transparency, biocompatibility, high flexibility, and physical and chemical stability. In addition, PDMS modification and combination with other materials can expand its range of applications. For instance, the ability to perform superhydrophobic coating allows for the manufacture of lenses. However, many of these processes are complex and expensive. One of the most promising modifications, which consists of the development of an interchangeable coating, capable of changing its optical characteristics according to some stimuli, has been underexplored. Thus, we report an experimental study of the mechanical and optical properties and wettability of pure PDMS and of two PDMS composites with the addition of 1% paraffin or beeswax using a gravity casting process. The composites' tensile strength and hardness were lower when compared with pure PDMS. However, the contact angle was increased, reaching the highest values when using the paraffin additive. Additionally, these composites have shown interesting results for the spectrophotometry tests, i.e., the material changed its optical characteristics when heated, going from opaque at room temperature to transparent, with transmittance around 75%, at 70 °C. As a result, these materials have great potential for use in smart devices, such as sensors, due to its ability to change its transparency at high temperatures.

**Keywords:** polydimethylsiloxane (PDMS); PDMS composites; interchangeable transparency; beeswax; paraffin

## 1. Introduction

Nowadays, polymers and elastomers have been attracting the attention of many researchers with its performance in the daily and environmental life of the planet due to their wide range of chemical and physical properties and excellent characteristics such as flexibility and corrosion resistance [1]. Among polymers, there is an increasing interest in the study of polydimethylsiloxane (PDMS) for applications such as mechanical and civil engineering, electronic devices, and in biomedical fields [2–7]. Within these areas, applications were reported such as water/oil and gas filtration membranes [8–10], sensors [11–13], lubricants [14], sealing agents [15], blood analogues [16–18], and also for microfluidic devices [19–23]. Recently, there has been a significant growing interest in microelectromechanical systems (MEMS) and microfluidic and optical devices [24].

This wide range of applications is justified by the desirable properties presented by this polymer, PDMS, a silicone based on organic polymers that is non-toxic, non-flammable,

and known as a material for which processing is simple with good repeatability and low cost. It is also an optically transparent material [11,25], biocompatible [26–28], highly flexible [29,30], waterproof [31], viscoelastic, and chemically and thermally stable [32–34].

Siloxane PDMS has been widely used in superhydrophobic coatings [35–37], consisting of surfaces with low surface energy, and has been a way to improve anti-fog, water-repellent [38], self-cleaning [39], anti-freeze [40], and anti-fouling [41] performance. Superhydrophobic coatings have recently shown promising anti-freeze properties by making water droplets slide over the air layer trapped in the nanoparticles with decreased adhesion, being thrown off the surface before freezing. Li et al. reported a composite pervaporation membrane prepared by coating a layer of SiO2/PDMS onto the hollow surface of the polyvinylidene fluoride (PVDF) membrane. The hollow fiber membranes showed improvement in the mechanical properties, and the surface shows good permeation of the phenol over the water, since the contact angle (CA) increased significantly by almost 50% [42]. Pakzad et al. used PDMS and beeswax to improve the surface of modified silicon nanoparticles. Both surfaces improved superhydrophobicity and corrosion resistance, and the surface modified with PDMS showed better durability against acidic solutions. Despite the positive results, they noted a complex manufacturing process, with difficulty of dispersal in the PDMS matrix [43,44].

Besides the applications involving superhydrophobic characteristics, PDMS has been applied in the last few decades in optical components due to its high transmittance and possibility of achieving a refractive index (RI) near the natural human lens, which is about 1.433. For instance, this allows for the use of PDMS in cataract surgery as a substitute for natural human lenses. The elastomer showed good results, as it can prevent a secondary loss of vision caused by the migration of lens epithelial cells (SCF) left behind during surgery [35]. Riehle et al. reported an RI between 1.401 and 1.403 for unmodified PDMS, and 1.4346 for a compound of PDMS and polysiloxane-urea-elastomers (PSU) [3]. Cruz-Felix et al. reported an RI between 1.408 and 1.445 for PDMS with different curing processes and showed a decrease in transmittance with an increase in curing temperature [45].

Other optical applications are in the microfluidics field, human and bionic lenses [3,46], micro-lenses [47], and photovoltaic panels [38,48], among others. Syafiq et al. produced a hybrid coating with PDMS and (3-Aminopropyl) triethoxysilane (APTES) by a simple immersion process, resulting in a waxy superhydrophobic surface with a contact angle above 100° and a transmittance above 90% in the visible region of the light spectrum, showing excellent transparency, self-cleaning, and anti-fog properties in indoor and outdoor environments [49]. Park et al. also manufactured a laminated film, composed of graphene, paraffin, and PDMS, with the same transmittance characteristics, and pointed this out as a promising technology for photovoltaic panels [25]. However, both authors used complex processes, involving chemical, thermal, or even electrical modifications during the manufacturing of the films. Additionally, a new trend is the development of materials capable of modifying their optical characteristics according to the received stimuli [50].

Ultimately, a subject was manufactured by the homogeneous dispersion of paraffin molecules in a PDMS matrix that significantly improved the thermal and optical properties of the compound and proposed the usage of the compound as a visual alert for the excess of temperature. This can reduce undesirable discomfort in wearable devices, generated when the soft substrate that is in contact with the skin was heated by the light emitting diode during the measurement process due to the limited heat dissipation capacity [6].

Accordingly, we report an experimental study for pure PDMS and for two PDMS composites with the addition of waxes: one using paraffin and the other using beeswax, both with low cost and wide availability. In the manufacturing process, instead of applying complex and costly chemical modification processes, additives were only inserted during a simple gravity casting technique. These latter choices were made based on a study carried out by Shi et al., where they compared several processes and concluded that the method used in the present study was simpler, faster, and less expensive [6].

To the best of our knowledge, no work has evaluated such a wide range of mechanical and optical properties for the composites used in the present study. Hence, the composites investigated in this study were analyzed by taking into account several mechanical, optical, and wettability properties. Additionally, the change in transmittance when the compounds were heated was investigated, and the results have shown that these materials have the possibility of being used in lenses and smart devices, which is an important new application.

## 2. Materials and Methods

### 2.1. Sample Manufacturing

The PDMS used to build the samples was the silicon Sylgard 184. The preparation of the material began with the mixture of the base material with its cure agent using a 10:1 ratio, as recommended.

Samples were performed with pure PDMS (0%), and with the addition of 1% paraffine (1%_P) or 1% beeswax (1%_B). The paraffin used was from simple candles, scraped with a knife. The beeswax was obtained from a local producer.

A gravity casting process was used. The base polymer was weighed for each specimen of pure PDMS used for tensile, spectrophotometry, and wettability tests, 5.45 g of base polymer and 0.54 g of curing agent were weighed. In order to obtain a homogeneous mixture, a small metallic spatula was used, making slow clockwise and counter-clockwise spins to avoid the formation of a large number of bubbles in the material. After that, the material went into a vacuum desiccator for 20 min to remove bubbles that were formed (degas process). In the sequence, the material was placed in the moulds and went again into the desiccator for 5 min before the curing process. The samples were cured at room temperature, approximately 25 °C, for, at least, 48 h.

As the paraffin and beeswax do not mix with PDMS at room temperature, it was necessary to heat the base elastomer while the mixture, with 1% of wax addition, was being made. The optimal temperature for mixing was 60 °C. This process is shown in Figure 1. In this case, for each specimen used for tensile, spectrophotometry, and wettability tests, 5.45 g of base polymer, 0.05 g of paraffin or beeswax, and 0.54 g of curing agent were weighed.

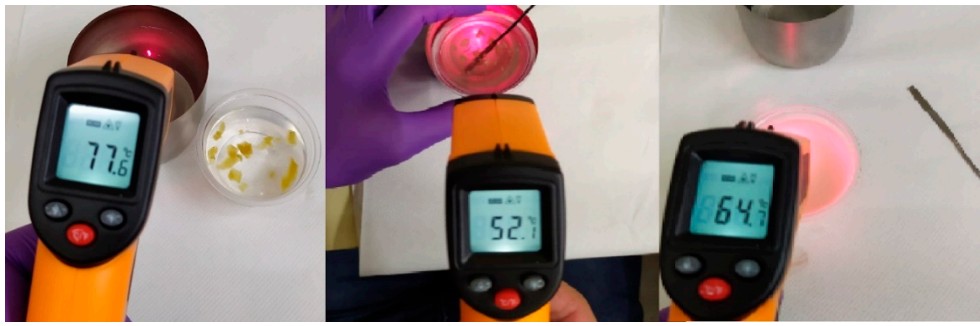

**Figure 1.** Procedure for mixing PDMS with 1% of beeswax.

After mixing, the cure agent was added; the mixture was homogenized with a metallic spatula, went into the desiccator for 20 min, and was placed into the moulds and placed in the desiccator for an additional 5 min.

Aluminum moulds were used to generate two different geometries used for each test. Firstly, for the tensile, spectrophotometry, and wettability tests, an aluminum mould was used, and the geometry was chosen following ASTM D412 [51], with a thickness equal to 2 mm.

Secondly, for the hardness test, samples were built in an aluminum rectangular mold with dimensions of 34 mm × 20 mm × 6 mm with rounded corners (3 mm radius). For each sample, 3.64 g of base polymer and 0.36 g of curing agent were weighed and, when mixing with paraffin or beeswax, the mass used for these additives was 0.04 g.

Figure 2 summarizes the whole method used to make the samples and the tests.

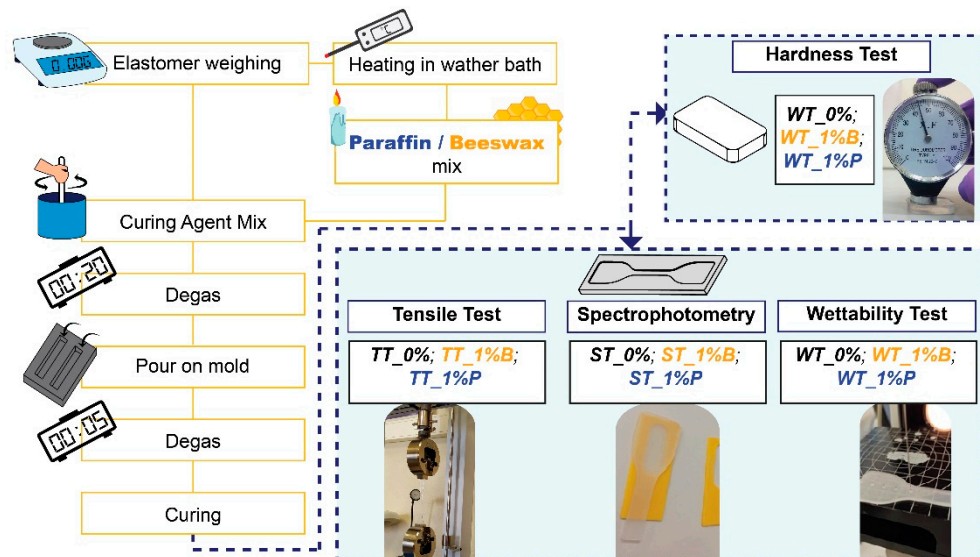

**Figure 2.** Sequence of steps for producing pure PDMS and composite samples, and the tests performed.

### 2.2. Tensile Test

Uniaxial tensile tests were carried out for 5 specimens of each sample using a universal tester machine, brand SHIMADZU, with a maximum capacity of 10 tonnes, and Trapezium X, version 1.5.1. software. The test was performed according to the ASTM D412 standard [51].

To perform the test, an initial setup of the machine was necessary, being configured with a pre-test with a velocity of 5 mm/min until achieving a pre-load of 1 Newton, and from this point, the test was set up for a velocity of 500 mm/min until the rupture of the sample. The distance between the grips was adjusted for 55 mm. For a better fixation of the samples, four thin metal plates with fine particle size sandpaper attached to the surface were used. These plates were important for avoiding the samples slipping during the test.

### 2.3. Spectrophotometry Test

Optical properties of materials can be determined by spectrophotometry. PDMS shows a transparent aspect in its pure state before being mixed with other materials. It is possible to see with the naked eye that the presence of a small percentage of beeswax gives it a yellowish translucent aspect. With the aim of quantitatively measuring the interference with the material's optical properties made by paraffin and beeswax, spectrophotometry tests were performed.

To execute the test, we used a spectrophotometer machine, brand SHIMADZU model UV-2600. Additionally, for this test, one specimen of each group was used. The test was performed after the samples were subjected to the tensile test. For that reason, the samples were placed in their larger parts between the two opaque plates with orifices, to guarantee that the signal emitted by the equipment passed only through the area of interest for the analysis.

The wavelength range set up for the equipment was between 200 μm and 800 μm. The measurements of transmittance were made at room temperature for all the materials and also with heated samples for the composites that had beeswax or paraffin in their composition.

To execute the heating of samples, a magnetic mixer was used with the samples emerged in water, raising the temperature until 80 °C. After a few minutes of heating, the samples were removed from the water, dried, and placed in the spectrophotometer.

The test was carried out at approximately 70 °C as a consequence of the thermal losses during the time between the removal from water and the positioning the samples in the equipment.

*2.4. Wettability Test*

A wettability test was performed according to the static sessile drop method. The samples used in this test were the specimens previously analyzed in the tensile tests. It was executed at room temperature, approximately 25 °C, using Data Physics brand Contact Angle System OCA equipment, and distilled water was used as the solvent.

Two specimens for each group were placed above the surface of the equipment and parameters such as focus, luminous intensity, and drop deposition were adjusted. Each of the drops was to stay 5 mm of distance away from each other, and the measurement was read five times. Figure 3 shows the positioning of the water drops on the samples with beeswax in the test.

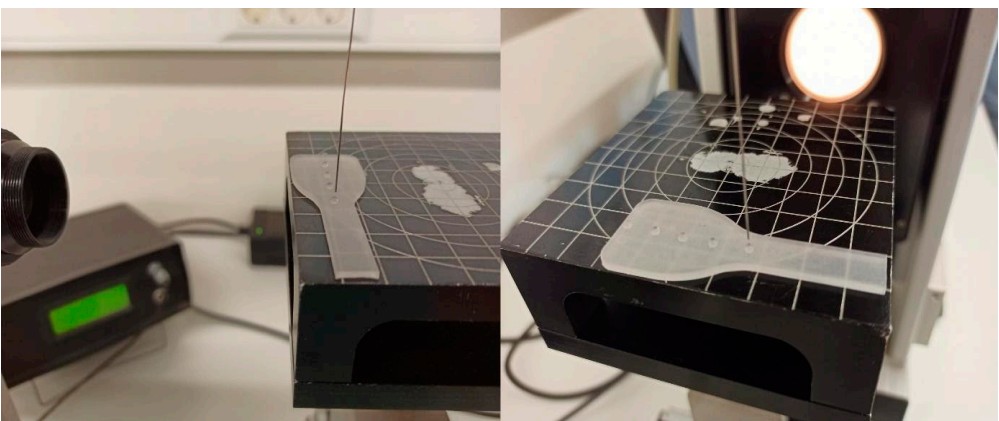

**Figure 3.** Positioning of the drops during the wettability test.

Note that the drops were displayed and recognized by the machine's software, SCA 20- Software for OCA and PCA, where the contact angle was automatically recognized and calculated.

*2.5. Hardness Test*

The measurements for hardness were carried out in an analogical portable durometer type Shore A according to ASTM D2240 [52].

To measure the hardness, two specimens of each sample were supported on a table with the flattest side of the samples being chosen for the test. Measurements at five points were made at approximately 18 °C.

**3. Results and Discussions**

*3.1. Tensile Test*

The graphs of engineering stress versus engineering strains are shown in Figure 4 for each of the samples and for the arithmetic means of the results. As the values obtained were dependent on the test parameters, and small variations in the positioning of the specimens could generate major changes in the results, it was decided to remove, in each of the groups of samples, the curves of the specimens with the greatest discrepancy. Thus, the results and mean values refer to four samples per group.

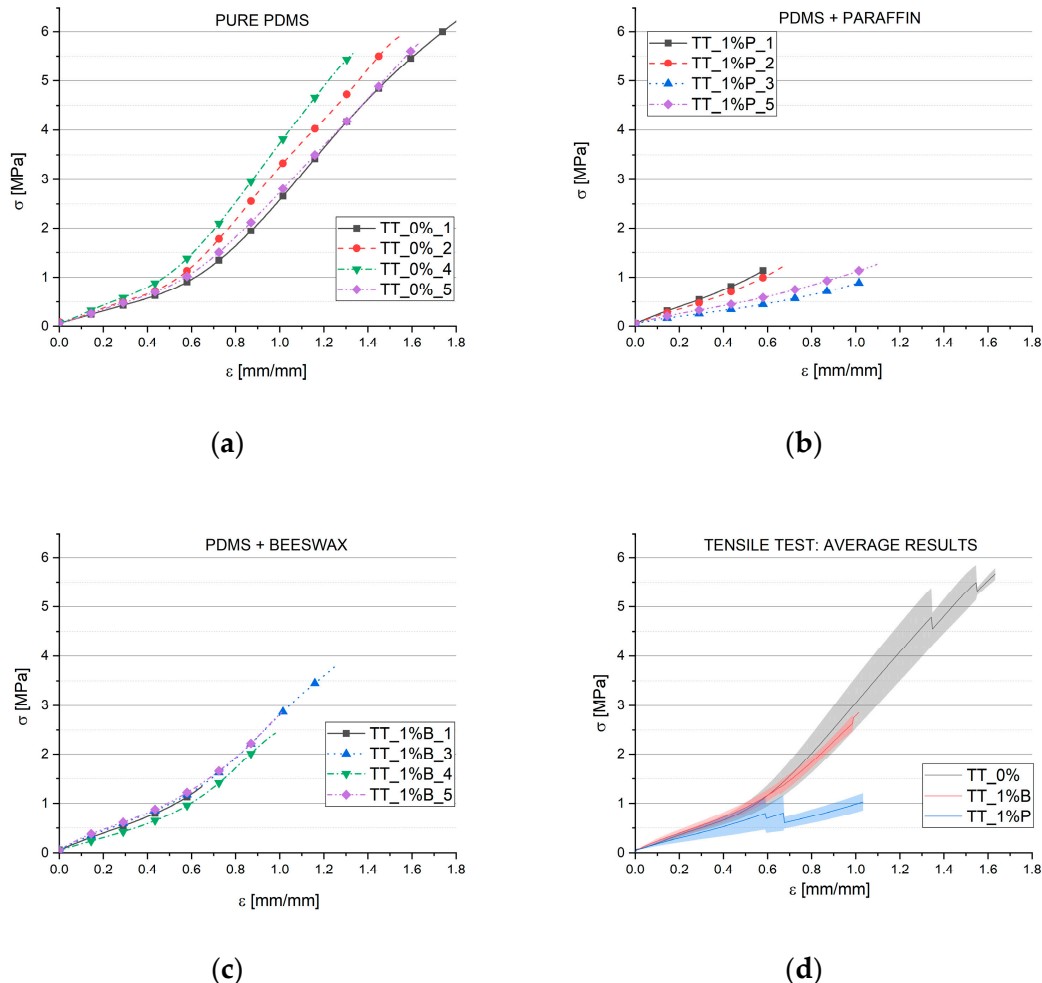

**Figure 4.** Engineering stress versus engineering strains for (**a**) pure PDMS (TT_0%); (**b**) PDMS with paraffin (TT_1%P); (**c**) PDMS with beeswax (TT_1%B); and (**d**) mean results and standard deviation.

The standard deviation related to the maximum stress of the pure elastomer samples was equivalent to approximately 10% of the obtained value, which corresponds to a deviation equivalent to the ones reported in the literature [24]. The biggest deviations were presented by the beeswax samples, as this material was obtained from a local supplier. Hence, further analyses should be done to evaluate this material's composition and how it influences the deviation of the results.

Analyzing the graphs for each of the samples, a typical behaviour of hyperplastic materials was observed, with an approximately linear region until high values for strain, around 40% [24]. In this region, the curves exhibited a similar behavior following the addition of a small amount of paraffin (TT_1%P) and beeswax (TT_1%B), which did not show a great change in the rigidity in comparison with the pure material (TT_0%).

However, achieving the non-linear region of the curve and, posteriorly, the fracture of the samples, the addition of paraffin and beeswax reduced the maximum supported tension (Table 1). The paraffin and the beeswax constituted new phases inside the PDMS matrix, although such phases could have performed as stress raisers rather than reinforcements. Thus, reaching values for strain around 1 mm/mm, those stress raisers contributed to a premature fracture of the material.

**Table 1.** Values of maximum stress and strains for pure PDMS and PDMS + paraffin and beeswax.

|  | TT_0% | TT_1%B | TT_1%P |
|---|---|---|---|
| $\sigma_{\text{máx.}}$ | $5.669 \pm 0.435$ | $2.852 \pm 1.016$ | $1.031 \pm 0.158$ |
| $\varepsilon_{\text{máx.}}$ | $1.633 \pm 0.245$ | $1.014 \pm 0.248$ | $1.033 \pm 0.256$ |

This behavior was already shown in the literature on studies with PDMS and paraffin ratios between 5 and 35%. For the addition of 5% paraffin, the results reported for maximum tensile stress were around 2 MPa, a value inferior than the that for pure PDMS [29]. The values expected from pure elastomer are around 5 MPa [24]. Pure beeswax and paraffin have a maximum stress of around 0.75 and 3.25 MPa, respectively [53].

*3.2. Spectrophotometry Test*

Pure PDMS is a material with high transparency, allowing for great light propagation. With the addition of small amounts of beeswax or paraffin, it is known that the transparency of PDMS can be changed [50], making the composite opaque. Pre-tests demonstrate that above 42 °C, the composite starts to become translucent, and after 64 °C, the appearance was very similar to that of pure PDMS, as shown in Figure 5.

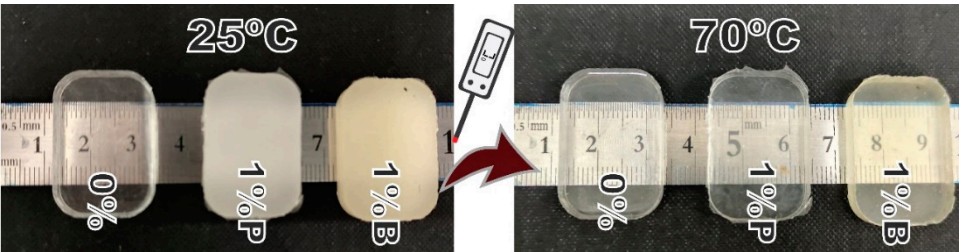

**Figure 5.** Change in the samples' transparency after the heating process.

For all spectrophotometry samples, the transmittance increased rapidly until reaching the visible wave spectrum, between 380 and 740 nm, as shown in Figure 6. At this level, the biggest differences were observed between the evaluated groups. Besides, in this region, the pure PDMS had a transmittance of around 75%, and this value remained approximately constant for the different wavelengths. In the studied composites at room temperature, the transmittance was reduced in comparison with the elastomer, and increased as the wavelength increased.

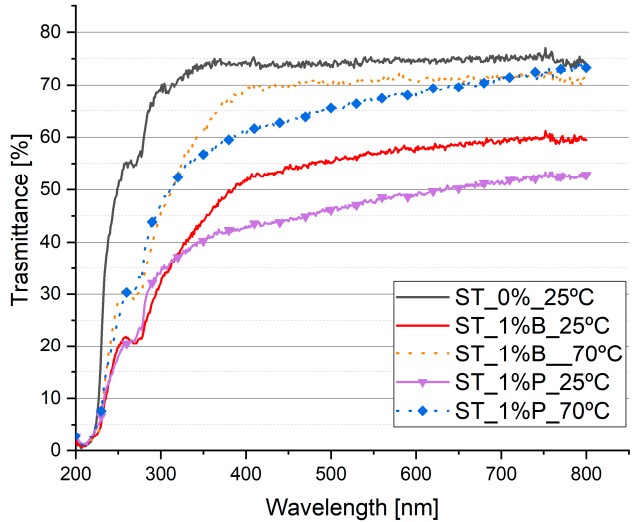

**Figure 6.** Wavelength versus transmittance for different temperatures.

However, for materials evaluated at 70 °C, the behavior became close to that of pure PDMS. For beeswax, after 400 nm wavelengths, the transmittance value was constantly near 70%. Paraffin showed an increasing transmittance, and an intersection of its curve with the pure PDMS' curve occurred for high wavelengths.

Upon returning to room temperature, the composites become opaque and showed cloudy appearance. This reversible behavior, with major changes due to thermal stimuli, can be justified by phase changes that occur after reaching the melting temperature of the additives. Below the melting temperature, the additives constituted a non-homogeneous solid phase that caused a great dispersion of light; when temperatures between 60 °C and 70 °C were reached, this phase melted and merged with the elastomer, producing a new homogeneous phase and increasing transmission [29].

References pointed out that in films with 2 mm thickness with the addition of 10% paraffin, the transmittance values are around 70% when analyzed at 20°C, and above 90% at 80 °C, which are values consistent with those obtained in this work [29].

### 3.3. Wettability Test

The results for different measurement points, shown in Table 2, presented a low standard deviation, which demonstrated the regularity of the surfaces in the different regions analyzed. Therefore, in all analyses, it was possible to observe the effect of the additives (paraffin and beeswax) on the roughness of the surface. This effect must be considered along with the surface roughness of the mold in which the material was produced.

**Table 2.** Values for the contact angle obtained for pure PDMS and PDMS with paraffin and beeswax.

| | Contact Angle | | |
|---|---|---|---|
| | Pure PDMS | Paraffin | Beeswax |
| **Measurement Point** | WT_0% | WT_1%P | WT_1%B |
| 1 | 120.8° | 135.8° | 135.0° |
| 2 | 116.9° | 145.3° | 120.6° |
| 3 | 122.3° | 143.6° | 133.6° |
| 4 | 120.4° | 141.4° | 127.3° |
| 5 | 115.8° | 143.5° | 130.2° |
| Arithmetic Mean | 119.2° | 141.9° | 129.3° |
| Standard Deviation | 2.8° | 3.7° | 5.7° |

The molds used in this work were made of an aluminum alloy, and had surface roughness (Ra) equal to 0.173 μm, allowing samples to be manufactured with close or superior roughness. Studies that explored PDMS combinations in order to create materials with a contact angle around 150°, showed Ra between 0.077 and 0.1285 μm.

For all the analyzed samples, the material exhibited hydrophobic behavior, with contact angles around 119.2° and 141.9°, for pure PDMS and PDMS with beeswax, respectively. Besides, the additives raised the contact angle, which caused the composite to behave similarly to a superhydrophobic material (with a contact angle higher than 150°). Those changes could have been generated by the reduction of the roughness and superficial energy, two of the main essential parameters for reducing the wettability of a material [44].

Furthermore, the beeswax had characteristics that made it an additive for manufacture superhydrophobic materials [54]. Regarding the paraffin, previous references have already demonstrated that its combination with PDMS allows for the development of materials that can reach contact angles of around 150°, likewise, in this work [44]. However, reported works have used additive ratios greater than 5%. In this way, this work proved that high contact angles can be reached using smaller amounts of paraffin.

Figure 7 shows the shapes of the drops and the contact angles for some of the analyzed points of the sample. In all the pictures, it is possible to observe the shape of the drops as essentially spherical, characteristic of hydrophobic materials.

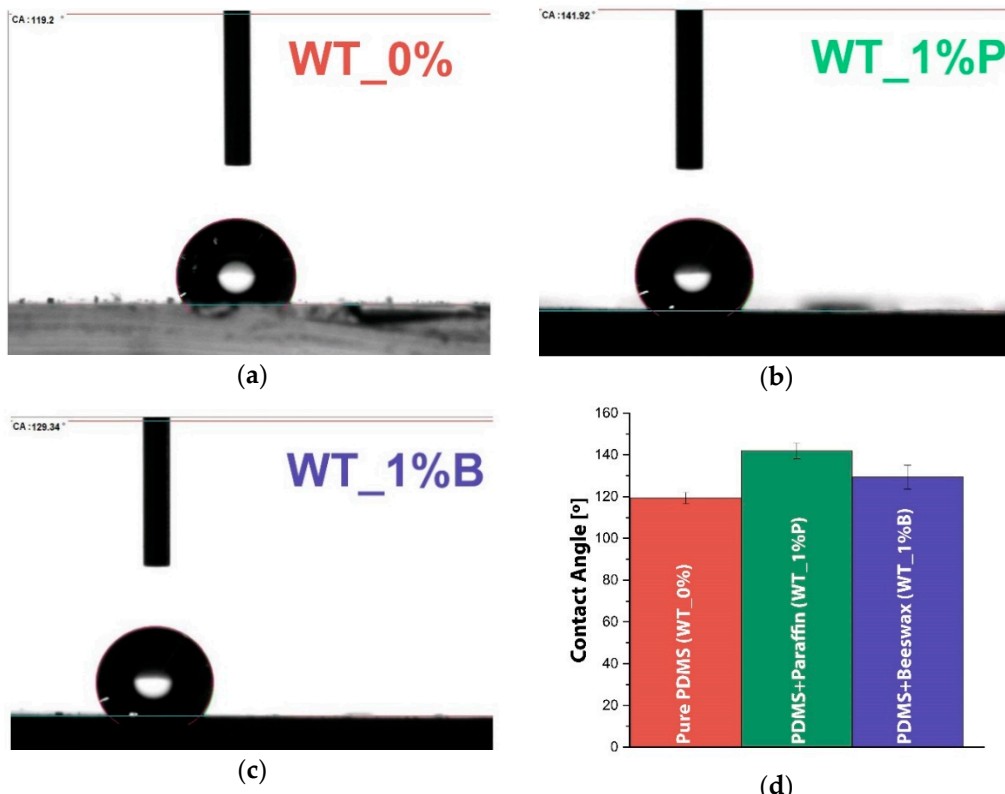

**Figure 7.** Obtained results of the wettability test for (**a**) PDMS 10:1, (**b**) paraffin, (**c**) beeswax, (**d**) mean values.

### 3.4. Hardness Test

As a result of the hardness test (Figure 8), the hardness average obtained for pure PDMS was 41.7 ± 0.95 Shore A, a value close to that declared by the manufacturer, which is 44 Shore A [55].

Regarding the other materials, the results showed that the addition of both paraffin or beeswax leads to a material with a more viscoelastic behavior at room temperature. For PDMS with paraffin, a mean hardness value of 33.2 ± 1.03 Shore A was obtained, and for PDMS with beeswax, 28.0 ± 1.05 Shore A, values 20.38% and 32.85% smaller than the pure PDMS, respectively.

This behavior was also presented in other past works where they have studied composites with a polymeric matrix and wax addition. In a low-density polyethylene (LDPE) matrix, the hardness of the pure material was 55 Shore A, and decreased to 51 Shore A when 2% of paraffin was added [56].

When adding wax, a reduction in tensile strength occurs, and a reduction in hardness was also expected, since these two properties are related [44]. Both mechanical properties are highly dependent on the network structure, mainly on the crosslink density and type [43]. As paraffin and beeswax were inserted, structural changes and weak interactions between PDMS chains can reduce the maximum stress and hardness reached [43,57].

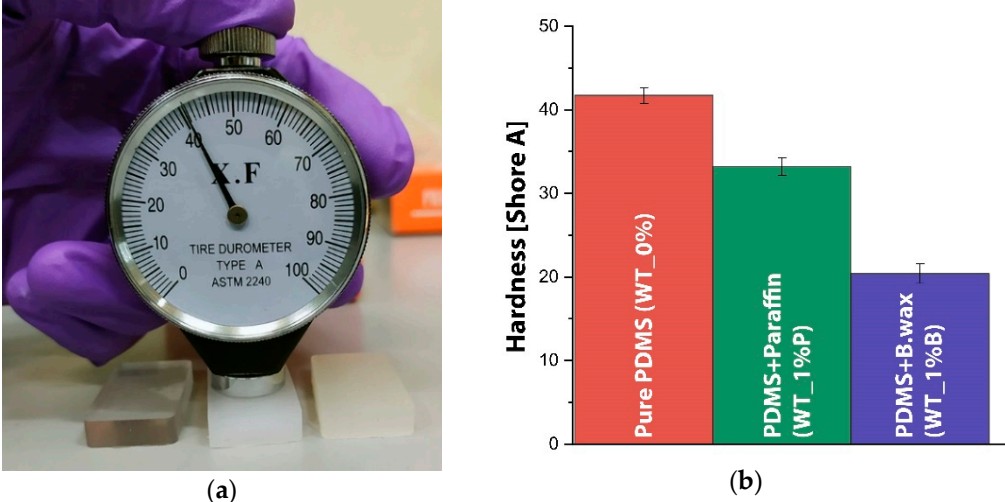

(a)                                         (b)

**Figure 8.** (**a**) Measurement with Shore A durometer; (**b**) results for hardness Shore A for each of the samples.

## 4. Conclusions

We have investigated the effect of wax addition on the PDMS properties. Extensive tests were conducted according to ASTM D412 and ASTM D2240 for tensile and hardness characterization. Additionally, wettability and transmittance evaluations were performed. This wide range of analyses using paraffin and beeswax composites have been barely explored in the literature, which makes the results obtained in this study extremely promising. Thus, the combination of all the properties summarized in Table 3, allows the application of these materials where all of these properties are required at the same time.

**Table 3.** Summary of the influence of wax addition on PDMS properties.

| | $\sigma_{máx.}$ [MPa] | $\varepsilon_{máx.}$ [mm/mm] | Hardness [Shore A] | Transmittance (Visible Region) [%] | Contact Angle [°] |
|---|---|---|---|---|---|
| 0% | $5.95 \pm 0.44$ | $1.61 \pm 0.25$ | $41.70 \pm 0.95$ | 74.37–75.45 | $119.24 \pm 2.76$ |
| 1%P | $2.60 \pm 1.02$ | $0.97 \pm 0.25$ | $33.20 \pm 1.03$ | 42.30–52.05 (25 °C) 59.51–72.44 (70 °C) | $141.92 \pm 3.69$ |
| 1%B | $1.13 \pm 0.16$ | $0.84 \pm 0.26$ | $28.00 \pm 1.05$ | 49.83–60.07 (25 °C) 67.63–71.25 (70 °C) | $129.34 \pm 5.73$ |

The results summarized in Table 3 show that the mechanical properties of the composites decreased when compared with pure PDMS. Acting as stress raisers, paraffin insertion reduced the maximum stress by 56.25%; for the beeswax insertion, this reduction was about 80.95%. The hardness reduction for paraffin and beeswax was around 20.38% and 32.85%, respectively. In contrast, the contact angle was increased, reaching the highest values (141.92°) when the paraffin additive was used. This value classifies this composite as hydrophobic, but with a behavior close to superhydrophobic materials. Hence, this composite shows a great potential for use as a cheap and simple method for water-repellent and anti-fog applications.

Spectrophotometry tests have shown interesting and unique results. At room temperature at the beginning of the visible region, the composites showed low transmittance, which tended to increase when the wavelength increased. Nevertheless, the values always remained below those of pure PDMS. In addition, the addition of paraffin and wax reduced the transmittance (wavelengths around 700 nm) by 20.38% and 31.02%, respectively. However, at a temperature of 70 °C, the material remained transparent in the extreme regions of visible light. As a result, these materials have a great potential for use in smart devices such as sensors capable of, by the change in passage of light, alerting the user when high



temperatures are reached. By adding paraffin, the transmittance grows approximately 22.74% from 380 nm to 700 nm, this is another potential application for manufacturing lenses and apparatus capable of filtering light at different wavelengths.

**Author Contributions:** Conceptualization, J.R. and R.L.; methodology, A.S.; software, F.S.; validation, F.S., R.A. and V.N.; formal analysis, R.L.; investigation, E.G.; resources, J.R.; writing—original draft preparation, R.A.; writing—review and editing, V.N.; visualization, A.S.; supervision, J.R.; funding acquisition, R.L. All authors have read and agreed to the published version of the manuscript.

**Funding:** This research was partially funded by Portuguese national funds of FCT/MCTES (PID-DAC) through the base funding from the following research units: UIDB/00690/2020 (CIMO) and UIDB/04077/2020 (MEtRICs). The authors are also grateful for the funding of FCT through the projects NORTE-01-0145-FEDER-029394, NORTE-01-0145-FEDER-030171, funded by COMPETE2020, NORTE2020, PORTUGAL2020, and FEDER.

**Institutional Review Board Statement:** Not applicable.

**Informed Consent Statement:** Not applicable.

**Conflicts of Interest:** The authors declare no conflict of interest.

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
