# Peer review of "Composite Material of PDMS with Interchangeable Transmittance: Study of Optical, Mechanical Properties and Wettability"

_jcs, doi:10.3390/jcs5040110_

Round 1

Reviewer 1 Report

  1. The authors affiliation should be completed conform the journal requirements.
  2. There are several articles published starting from PDMS and with paraffin, paraffin oil or wax. I would like to ask the authors to introduce the most important findings from these articles to the introduction of the present manuscript.
  3. What is the novelty of the present manuscript, please emphasize in the introduction and conclusions.
  4. A bit more detailed “sample manufacturing” would be welcomed.
  5. I would like to ask for an explanation for lower contact angle values for the presented composites with beeswax and paraffin comparing to those one mentioned for the literature?
  6. The majority of the references are not visible on the text, there is an error: “Error! Reference source not found”. Please correct this issue.

Author Response

Firstly, we would like to acknowledge the reviewer for spending his/her valuable time reading our manuscript and for the relevant comments and suggestions. Please note that the changes made in this revised version of the manuscript are highlighted in yellow.

  1. The authors affiliation should be completed conform the journal requirements.

We would like to thank the reviewer for this observation. This information was now added according to the journal requirements.

  1. There are several articles published starting from PDMS and with paraffin, paraffin oil or wax. I would like to ask the authors to introduce the most important findings from these articles to the introduction of the present manuscript.

Thank you very much for your suggestion. As suggested, more details about the results of these studies using PDMS and paraffin were added in the introduction and discussion.

  1. What is the novelty of the present manuscript, please emphasize in the introduction and conclusions.

Thanks a lot for the observations that will help to improve the manuscript. Several changes were made in order to emphasize in the introduction and conclusion that very few studies were made to characterize different properties of this material with interchangeable transmittance. Another novelty that was included in the manuscript, was by using a very small proportion of additives we have found significant changes in the optical properties of the proposed materials. In addition, a low cost and simple manufacturing process were used in this work.

  1. A bit more detailed “sample manufacturing” would be welcomed.

We are very grateful for your suggestion. We have included more information about this part of the methodology, specifically quantitative details, such as reagent weights; time for each process steps; fluid used in the wettability tests; the number of specimens used and their dimensions.

  1. I would like to ask for an explanation for lower contact angle values for the presented composites with beeswax and paraffin comparing to those one mentioned for the literature?

Thank you very much for your question and we are very grateful for the opportunity to explain to you the possible reasons why this could have happened. Firstly, roughness tests were conducted to evaluate how the moulds can interfere with the PDMS surface. Considering that this material is able to replicate even microfluidic channels, its roughness should be close or slightly higher than the one showed by the aluminium moulds. Thus, this test has shown that, due to the mould manufacturing process, the roughness was higher than the ones reported in the literature. In addition, and more importantly, the used additive ratio was much smaller than the one found in the literature. These main explain the difference between our values and the contact angle values found in the literature.

  1. The majority of the references are not visible on the text, there is an error: “Error! Reference source not found”. Please correct this issue.

Thank you very much for your observation. This error was solved and corrected.

Reviewer 2 Report

General comments

There are several papers mentioning the same composites. It is not clear what is the novelty/importance of this study. It is only a duplication, there is nothing new.

Sample and specimen have different meaning. In this research three samples were used. For each sample several specimens were moulded. Please, check through the manuscript.

The number of specimens must be clearly mentioned. The notation of the specimens must be explained before. P and B are used in figures without a previous explanation.

In Fig 4 some specimens have low values. This is not clear neither mentioned during the discussions. In some specimens (with beeswax and paraffin) the tensile stress is higher than the values of pure PDMS specimens.

The maximum/mean values in Table 1 are not according to the data in Fig 4.

The discussion must be improved. Are the stress and transmittance values according to published results? Do different paraffins and beeswaxes give similar results? This topic is never discussed.

Specific comments

Line 91 - “Accordingly, we report an experimental study of mechanical (hardness and tensile), optical and wettability properties for pure PDMS and for two PDMS’ composites with waxes addition: one using paraffin and the other using beeswax”

There are several researches mentioning the same composites. It is not clear what is the novelty of this study.

Line 104 - “The paraffin used was from simple candles, scraped with a knife. The beeswax was obtained with a local producer.”

The paraffin and beeswax were not analysed. Can the results be extrapolated for any of these materials?

Line 116 - “process is shown in Error! Reference source not found.”

Check…

Line 127 - “Error! Reference source not found. summarizes”

Check!

Line 169 - “wettability test.Error! Reference source not found.”

Check!

Line 181 - “The graphs of engineering stress versus engineering strains are shown for each of the samples and for the average results in Figure 4.”

Average has a broad meaning (mean, median, mode, …). “arithmetic mean” or “mean” are better options.

Please, inform readers about the number of the specimens used in the tests.

Line 189 - “the addition of paraffin and beeswax reduced the maximum supported tension”

This is not valid for the specimen TT_1%B_2 (see Fig 4c).

Line 190 - “tension (Error! Reference source not found.). The paraffin”

Check.

Line 196 - “Table 1. Values of maximun stress and strains for Pure PDMS and PDMS + Paraffin and Beeswax.”

What about the standard deviation?

Line 201 - “Pre-tests, shown in Error! Reference source not found., demonstrate that above 42ºC the composite starts to become translucent and, after 64ºC, the appearance was very similar to that of pure PDMS, as shown in Error! Reference source not found.”

Check through the manuscript for these missing references.

Line 239 - “Regarding the paraffin, previous references have already demonstrated that its combination with PDMS allows the development of materials that can reach contact angles around 150°, likewise this work.”

Please, add the references.

Line 253 - “Regarding the other materials, the results showed that, the addition of both paraffin or beeswax leads to a material with more viscoelastic behaviour at room temperature. For PDMS with paraffin, it was obtained a hardness average of 33.2±1.03 Shore A, and for PDMS with beeswax, 28.0±1.05 Shore A, values 20.38% and 32.85% smaller than the pure PDMS, respectively.”

Are the values according the previously published values? During the discussion, only the value declared by the manufacturer is mentioned.

Line 266 - “Table 3. Summary of the influence of wax addition on PDMS.”

Mean and standard deviation values….

Tables

Table 1 - Please, check this sample: TT1%C

Please, explain how the maximum stress can be 5.953±0.435 if just one specimen show value above 6 and one sample only reaches 3. The same happens with other samples. The mean values are not according the data in Fig 4.

Table 3 - Check the decimal notation.

Author Response

We would like to acknowledge the reviewer for spending his/her valuable time reading our manuscript and for the relevant comments and suggestions. Please note that the changes made in this revised version of the manuscript are highlighted in yellow.

  1. There are several papers mentioning the same composites. It is not clear what is the novelty/importance of this study. It is only a duplication, there is nothing new.

We respect the reviewer opinion and we have clarified the novelty of our work by making several changes in the manuscript. In this way, we have emphasized in the introduction and conclusion that very few studies were made to characterize different properties of this material with interchangeable transmittance. Another novelty that was included in the manuscript, was by using a very small proportion of additives we have found significant changes in the optical properties of the proposed materials. In addition, a low cost and simple manufacturing process were used in this work.

From our literature review, the majority of the studies only have analysed each additive separately (paraffin or beeswax) and do not evaluate ae wide range of properties as it is presented in our work. All these features are extremely relevant, as it may allow us the use this material for open-air applications.

  1. Sample and specimen have different meaning. In this research three samples were used. For each sample several specimens were moulded. Please, check through the manuscript.

We would like to thank the reviewer for the observation. The mistakes using the words “sample” and “specimens” were corrected throughout the manuscript.

  1. The number of specimens must be clearly mentioned. The notation of the specimens must be explained before. P and B are used in figures without a previous explanation.

Thanks a lot for this comment. The number of specimens and notation was inserted in the methodology section.

  1. In Fig 4 some specimens have low values. This is not clear neither mentioned during the discussions. In some specimens (with beeswax and paraffin) the tensile stress is higher than the values of pure PDMS specimens.

Thank you for your observation. We are very grateful for the opportunity to clarify the possible reasons why we have reached these results. As the specimens have low resistance and because they were tested at a high crosshead motion speed, small changes at the test procedure may have occurred due to incorrect alignment of the claws that can generate large discrepancies in the results. This might be one of the reasons for the difference between our results and the results found in the literature for pure PDMS. The standard deviation of the maximum stress is around 10% of its value when compared with the results performed by Johnston et al.  [24] (Johnston, 2014).

  1. The maximum/mean values in Table 1 are not according to the data in Fig 4.

We appreciate the reviewer for this observation. It happened because, in the analysis, the most discrepant curves were disregarded since they do not represent the standard behaviour of the material and, by mistake, we have included these curves in graphs of Fig. 4. These curves were precisely those with additives and present values higher than the ones shown for pure material. In this updated version, we have corrected this Figure and they are no longer shown in Figure 4 and, in the text.

  1. The discussion must be improved. Are the stress and transmittance values according to published results? Do different paraffins and beeswaxes give similar results? This topic is never discussed.

Thank you very much for your suggestions that will improve the quality of the manuscript. As a result, several improvements and corrections were made in Table 1 and in the discussion. The comparison between our results and the ones found in the literature were added to the discussion section.

Reviewer 3 Report

Dear Authors, Congratulation  to this paper.

I have only two remark.

1) contact angle
Please add the used solvent. I think it was water, but you do not write this. 
2) Formatting
In the pdf File are a lot Error! Reference source not found. Please check it in the version for publication.

BR 
The reviewer

Author Response

We would like to acknowledge the reviewer for spending his/her valuable time reading our manuscript and for the relevant comments and suggestions. Please note that the changes made in this revised version of the manuscript are highlighted in yellow.

  1. Contact angle: Please add the used solvent. I think it was water, but you do not write this.

We would like to thank the reviewer for the observation. The solvent used was pure water. This information was now added in the manuscript at the methodology section.

  1. Formatting: In the pdf File are a lot Error! Reference source not found. Please check it in the version for publication.

Thank you very much for your observation. These errors were solved and corrected.

Round 2

Reviewer 1 Report

Thank you very much for the updated version of the article. The quality of it is improved, but there are still some minor things to fix before the publication.

  1. Lines 90-103 – there are no references linked to the newly introduced part. Please complete with the appropriate references.
  2. What kind of approximation did you used for the contact angle measurements? Please mention in chapter 2.4.
  3. I would like to ask for the insertion of a small discussion part just before the conclusion part.
  4. In my version the error (“Error! Reference source not found”) is still exist in several places.

Author Response

Thank you very much for the updated version of the article. The quality of it is improved, but there are still some minor things to fix before the publication.

  1. Lines 90-103 – there are no references linked to the newly introduced part. Please complete with the appropriate references.

Thanks a lot for your comment and suggestion. I have added more references to the text, mainly from authors already mentioned in the other introductory parts.

  1. What kind of approximation did you used for the contact angle measurements? Please mention in chapter 2.4.

We thank you for this comment that has contributed to the addition of more detailed information in the text. The tests were conducted on a device (Contact Angle System OCA) coupled to a computer with the equipment's proprietary software (SCA 20- Software for OCA and PCA). Both equipment and software are distributed by the company Data Physics. Therefore, the analysis system automatically recognizes the images, fits a mathematical function to the drop shape and provides, as an output, only the respective contact angles.

Besides the addition of this information in the manuscript, we have also included the methodology adopted in the tests, i. e., "static sessile drop method".

Reviewer 2 Report

General comments

Analysing the two additives at same time cannot be considered a novelty. Authors argue that use a small quantity of additives and use a novel approach by inserting additives during the gravity casting technique. Therefore, previous techniques must be clearly described. Furthermore, the discussion of the results should consider the previous published results.

The authors didn’t consider some of my specific comments, neither in the answers nor in the revised manuscript.

Author Response

We deeply apologize for the mistakes during the first reviewing process. We had some problems with the visualizations of the comments on the online platform and we did not see the specific comments located at the end of the platform. We would like to thank you for all these comments and suggestions. The main suggested changes were already done after the first reviewing step, others were done on this new version of the manuscript. However, we have pointed out your previous comments and their respective answers. Note that the order of the comments was reorganized below.

Analysing the two additives at same time cannot be considered a novelty. Authors argue that use a small quantity of additives and use a novel approach by inserting additives during the gravity casting technique. Therefore, previous techniques must be clearly described. Furthermore, the discussion of the results should consider the previous published results.

The authors didn’t consider some of my specific comments, neither in the answers nor in the revised manuscript.

Previously comments:

  1. Line 91 - “Accordingly, we report an experimental study of mechanical (hardness and tensile), optical and wettability properties for pure PDMS and for two PDMS’ composites with waxes addition: one using paraffin and the other using beeswax” There are several researches mentioning the same composites. It is not clear what is the novelty of this study.

Thanks a lot for the observations, they will help to improve the manuscript. We had already added excerpts discussing the innovation presented. However, we understand the importance of clarifying these factors, as well as pointing out other references in the literature that justify them. In this way, more changes were done in the introduction and conclusion in order to clarify the novelty. Please note that despite several authors have already studied this composite, to the best of our knowledge no study has done such a wide range of measurements for the specific composites tested in this study.

  1. Line 104 - “The paraffin used was from simple candles, scraped with a knife. The beeswax was obtained with a local producer.” The paraffin and beeswax were not analysed. Can the results be extrapolated for any of these materials?

Most authors in this field only point out the supplier of the materials, for instance, the following works with paraffin [29, 11], and beeswax [44], present in the bibliography of the manuscript.

For this reason, we added a comparison with some other studies that have analysed the paraffin and beeswax separately, where the composition of these materials changes slightly. However, for future works, we intend to carry out detailed chemical analysis and characterization of those materials.

Reference errors:

  1. Line 116 - “process is shown in Error! Reference source not found.”
  2. Line 127 - “Error! Reference source not found. summarizes”
  3. Line 169 - “wettability test.Error! Reference source not found.”
  4. Line 190 - “tension (Error! Reference source not found.). The paraffin”
  5. Line 201 - “Pre-tests, shown in Error! Reference source not found., demonstrate that above 42ºC the composite starts to become translucent and, after 64ºC, the appearance was very similar to that of pure PDMS, as shown in Error! Reference source not found.”
  6. Check through the manuscript for these missing references.

We apologize again for this mistake. We believe that we have identified the origin of this mistake and hopefully, this mistake was solved.

Terminology mistakes: “average” instead of mean.

  1. Line 181 - “The graphs of engineering stress versus engineering strains are shown for each of the samples and for the average results in Figure 4.” Average has a broad meaning (mean, median, mode, …). “arithmetic mean” or “mean” are better options.
  2. Line 266 - “Table 3. Summary of the influence of wax addition on PDMS.” Mean and standard deviation values….

Thanks a lot for the observation that will help to improve the paper. We read through all the manuscript and changed the average for the corresponding terms.

  1. Please, inform readers about the number of the specimens used in the tests.

We would like to thank you for this comment. Quantitative information was added to the sample manufacturing (2.1) and tensile test (2.2) section. It was included the mass of polymers and additives used, time for each step, main dimensions, and the number of specimens.

Tensile Test Graphics and stresses.

  1. Line 189 - “the addition of paraffin and beeswax reduced the maximum supported tension”. This is not valid for the specimen TT_1%B_2 (see Fig 4c).
  2. Please, explain how the maximum stress can be 5.953±0.435 if just one specimen show value above 6 and one sample only reaches 3. The same happens with other samples. The mean values are not according the data in Fig 4.

Thanks for your comments. During de data analysis and treatment process, some of the specimen’s information were not considered due to the high deviation, probably caused during the tests. This information was added to the manuscript and the figure was updated to show only the plots considered during all the analysed processes.

  1. Line 196 - “Table 1. Values of maximun stress and strains for Pure PDMS and PDMS + Paraffin and Beeswax.” What about the standard deviation?

Thank you for your comment. The standard deviation was now discussed on the Results and compared with the literature.

  1. Line 239 - “Regarding the paraffin, previous references have already demonstrated that its combination with PDMS allows the development of materials that can reach contact angles around 150°, likewise this work.” Please, add the references.

We are grateful for this suggestion that has contributed to improving the understanding of the results. The references were properly added.

  1. Line 253 - “Regarding the other materials, the results showed that, the addition of both paraffin or beeswax leads to a material with more viscoelastic behaviour at room temperature. For PDMS with paraffin, it was obtained a hardness average of 33.2±1.03 Shore A, and for PDMS with beeswax, 28.0±1.05 Shore A, values 20.38% and 32.85% smaller than the pure PDMS, respectively.” Are the values according the previously published values? During the discussion, only the value declared by the manufacturer is mentioned.

Thank you for your question and suggestion that was important to improve the discussion of the results. We have clarified in the paper that we have found just a few studies to perform comparisons, as the majority of the studies found in the literature the composite has a high amount of additive. However, we have included a comparison with other composites with different matrix and the same additives, and we believe that may help to understand the effects of those additives.

  1. Table 1 - Please, check this sample: TT1%C

We would like to thank the reviewer for this observation. This sample’s name was changed from TT1%C (C means “cera”, the Portuguese word for wax) to TT1%B (B comes from beeswax).

  1. Table 3 - Check the decimal notation.

Thanks a lot for the observation, the decimal notation was checked and corrected.

Reviewer 3 Report

Now it is okay

Author Response

Thank you very much for accepting our manuscript.

Round 3

Reviewer 1 Report

The article can be published in present form